# Student Experiences and Changing Science Interest When Transitioning from K-12 to College

David E. Reed [1],*, Emily C. Kaplita [2], David A. McKenzie [3] and Rachel A. Jones [1]

1 Division of Science and Physical Education, University of Science and Arts of Oklahoma, Chickasha, OK 73018, USA; rjones@usao.edu
2 Department of Biology, Dickinson College, Carlisle, PA 17013, USA; eckaplita@gmail.com
3 Department of Biological Sciences, Emporia State University, Emporia, KS 66801, USA; dmckenz1@emporia.edu
* Correspondence: dreed@usao.edu

**Abstract:** Student attitude and involvement in the sciences may be positively or negatively influenced through both formal academic experiences and informal experiences outside the classroom. Researchers have reported that differences in science interest between genders begin early in a student's career and that attitudes towards a particular field of science can be correlated to achievement in that field. In this study, we approach the question of how attitudes towards science have been shaped using college-age students. Survey data from students in similar academic positions were employed to control for differences in cultural and academic progress. Results from a self-reflection survey indicated that general personal interest in both science as a process and field-specific content increased from elementary school through high school until entering college. Differences arose between self-identified genders in student experiences with science, both while in groups and when on their own. Female students had higher rates of participation and enjoyment with science in groups, while male students more frequently enjoyed science alone. Students, regardless of gender, rarely had negative experiences with science outside of the classroom. However, male students' interest in science surpassed female students' during high school. Declining interests in quantitative aspects of science (mathematics and statistics) were more frequently reported by female students and non-STEM majors during and before their college experience. Connecting student attitudes regarding science to their pre-college experiences with science early in their college career may be important to understanding how to best engage all genders, as well as non-STEM majors, in their college science courses.

**Keywords:** science interest; college transition; student experiences

## 1. Introduction

At both national and global levels, citizens are faced with many large-scale science-related issues; climate change, food and water resource shortages, and disease spread, to name only a few. It is vital for voting-age students and members of society to be educated in the science underlying these issues in order to make informed decisions regarding their mitigation and/or regulation as citizen voters [1]. Mitigation of climate change, mass pollution of water and land, water scarcity, and more all require some level of familiarity with underlying scientific concepts. In fact, science and the general public are more disconnected than ever [2], even as societies have grown due to the industrial and technological developments science has provided. It is imperative that STEM (science, technology, engineering, and mathematics) becomes a more significant part of our education system. Beyond training workers in the STEM fields, it is equally important to improve science literacy amongst all citizens in order to garner support for our societies' continued expansion in these fields.

Science literacy is a topic that has been the focus of much research, with ongoing improvements in science literacy being reported using pedagogy techniques such as student engagement [3], inquiry-based learning [4], laboratory activities [5], and targeted courses for non-science majors [6]. However, teacher-focused pedagogy is only one factor in improving science literacy among students. Student attitudes toward and interest in science also has a large influence on the scientific literacy of a student. With noting that the idea of gender is broad, complex, and can change, student gender has long been studied as one possible explanation for differences in literacy between male and female students [7–9]. In addition to student gender, the learning environment and a student's experiences with instructors also influence student attitude and achievement in science [10]. An instructor's understanding of pedagogy practices successful in STEM education and implementation allows them to tailor activities and lectures to mitigate potential negative attitudes towards science and may help reduce the gender gap in STEM fields [11,12]. Along with formal settings in the classroom, students form ideas about science through social interactions with peers [13] and in informal settings before starting school [14], with large impacts on student readiness for school and achievement in school.

The connection between a student's interest in science and achievement in science is important for younger students [15,16]. Maintaining and supporting student interest in science at all levels of education is important, as young people's attitudes towards science today have large implications for the rates of science literacy in the general population of the future [10]. Many studies in the past have focused on student interest in science focusing within the K-12 years [11,12,17–19] and during college [20–22], with less research on the transition from K-12 to college. However, a study by Tai, Liu, Maltese, and Fan [23] indicates the choice to pursue a career in the STEM fields may, in fact, occur later in high school or even early in a student's college career. For this reason, it is important to understand the reasons behind students entering STEM fields at the appropriate time in their education.

Gender-based gaps in science literacy are a topic that has captured the attention of both education researchers and policy makers [24]. Promoting diversity of all kinds within STEM fields should be a priority for both academia and society as a whole. STEM retention rates are often a key statistic studied, along with the number of students graduating with specific majors at the college level [25]. However, gender disparities in selecting whether or not to major in STEM fields have roots in pre-college experiences, and more female than male students enter college planning to major in non-STEM fields [26]. Understanding the source of what appears to be a disinterest by female students is key to addressing the disparity between genders.

It is common for students to report science or math anxiety, with this anxiety stemming from a single negative event or experience. Studies indicate some of these anxious feelings may also be a function of gender [27,28]. A better understanding of these early experiences would help in preparing and instructing introductory level college classes because students often come to college with firm ideas regarding science, but these ideas are based only on a small number of potentially negative events. The work of Reed and Lyford [6] quantifies the potential positive change in student attitudes toward science after taking a single introductory science-for-non-majors course at the college level. Science anxiety can be difficult to overcome in just one course, and the number of required science classes for non-STEM majors is decreasing at many colleges. Having well-rounded students leaving college is a common goal of higher education, but for this to happen, positive experiences with science during college must overcome negative K-12 experiences.

Our aim of this work was to (1) Characterize differences in science interests that persist from K-12 into college between self-identified genders; (2) Assess the timing of negative and positive experiences in science in both formal and informal settings and determine how these experiences relate to current interest and understanding toward science.

## 2. Materials and Methods

While not without limitations [29], surveying students allows researchers to indirectly quantify student interest levels in a topic. By surveying students early in their college careers and asking them to report on interest levels in and experiences with science in the past, we are able to identify key points that stand out to students as being important in their current attitudes towards science. Unfortunately, student survey data can be biased in several ways, including how the survey was administered [30] or by cultural differences within the student population [31]. In order to limit sample bias, we targeted student populations at multiple kinds of postsecondary institutions across a single metropolitan area. By sampling multiple colleges, we also captured potential variation in the student population's propensity to apply and enroll in different types of colleges.

This study involved assessing the student's interest and understanding of science before college. A forty-question survey was designed to ask questions regarding the student's experiences with science in the classroom, with family and friends, on their own, interest in science over time, and demographic questions. The survey was given to course instructors at three universities, the University of Oklahoma (*n* = 249), the University of Science and Arts of Oklahoma (*n* = 60), and Rose State College (*n* = 47), all located in the Oklahoma City Metro Area. The surveys had 41 questions each and were given to 100 and 200-level science classes, including general physics, general chemistry, general biology, introduction to botany, and introduction to ecology, during the first two weeks of each institution's semester in the fall of 2016. Students were informed the survey was voluntary and their informed consent was asked for before starting the survey. A copy of the survey is availed as a Supplementary File. Courses were chosen to capture a wide variety of students across colleges, all with similar demographics and backgrounds in K-12 educational experiences. A total of 63% of students were from Oklahoma, and another 17% of students were from Texas. An invitation to participate in this study was sent to all STEM professors at the three schools sampled.

All surveys were collected from a total of 356 students. A small number of surveys (*n* = 5) that were not completed were flagged and not included in the study. The total compiled database was then separated by self-identified gender (*n* = 113 Male students and *n* = 238 Female students) and major (*n* = 231 STEM, *n* = 120 non-STEM) for analysis. The data were analyzed using equal variance *t*-tests to test statistical differences between genders or majors using a 95% confidence value. T Survey results were normalized by converting total number of responses to percentages. Results are shown in all figures for all respondents as a single group as well as gender and major subgroups, with non-statistical differences between subgroups shown as white hatched bars.

## 3. Results

The general trend among all students was that their interest in science as a process increased as a student aged (Figure 1A), as 48% of students were interested in the scientific process in elementary school and 68% in college. Male students started with lower interest in science (41%) compared to female students (51%) in elementary school (*p* = 0.02) but had a larger increase in interest and ended up equal to female students in college. STEM students started with a lower interest in science, but as they moved through school, their interest increased the most, from 45% to 82%. This differs from the non-STEM students, who decreased their interest in science from 54% to 41%. When asked about their interest in science content throughout time, there was a similar story reported by the students (Figure 1B). STEM students' interest in science content increased with age (46% to 86%), while non-STEM students' interest in science content was the same in elementary school as in college (54–55%), with a small peak in late high school (60%). Again, male student interest started smaller (44%) and increased (81%) compared to female students, who had a higher initial interest (51%) but did not gain interest as fast as male students and ended up at a lower level in college (73%). Elementary school interest in science content for all

students was the same as interest in science as a process (48%), but college-level interest in science content was higher at 76% ($p = 0.04$).

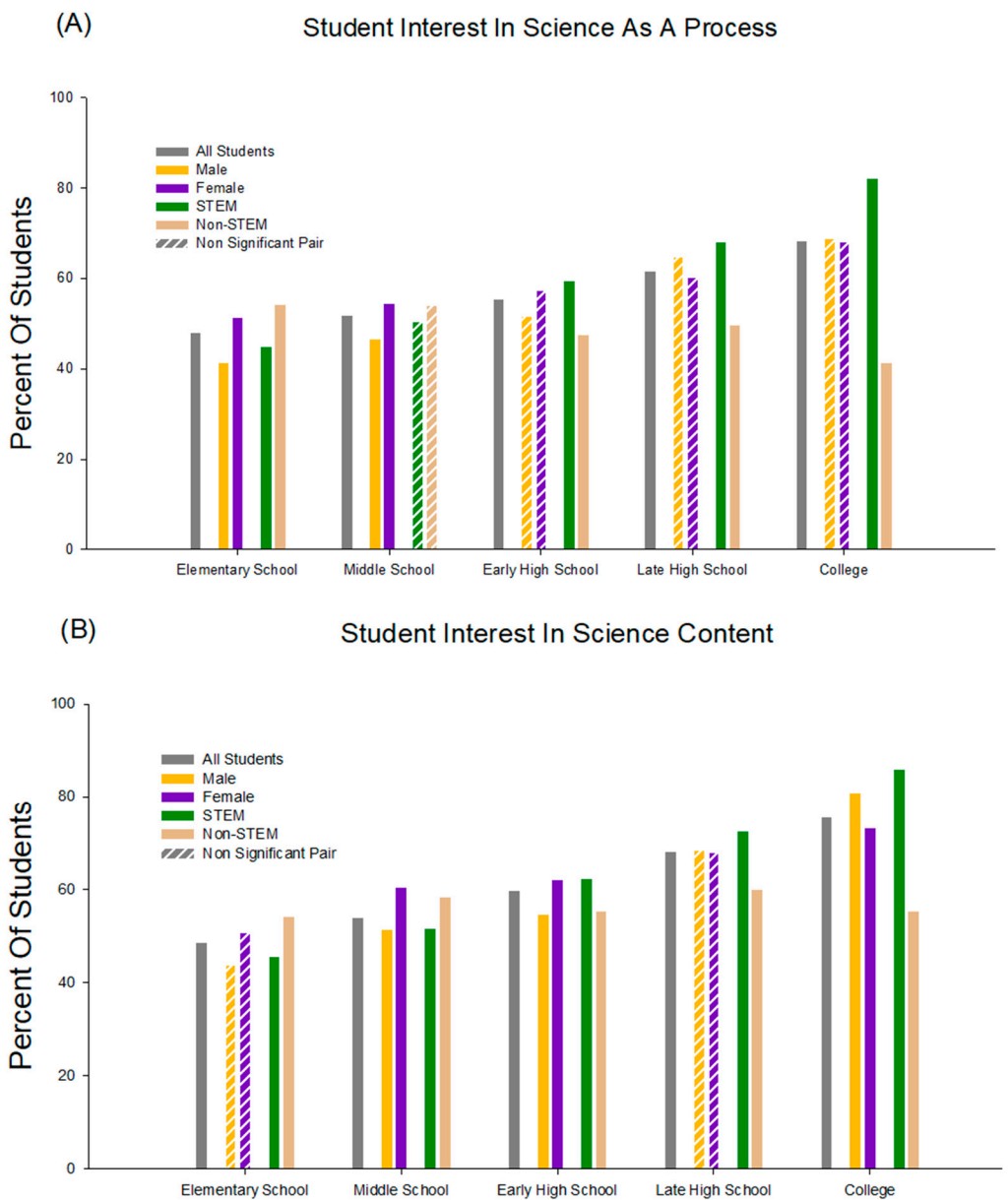

**Figure 1.** Student interest in science as a process (**A**) and science content (**B**) is shown. All students are shown (grey), together with the data split by self-identified gender (males/yellow and females/purple) and college major (STEM/green and non-STEM/tan). Non-significant differences between gender and major splits are shown for each period are shown as white hatched bars.

Before enrolling in college, students had a diverse set of experiences with science, with nearly every student having taken at least one science course (Figure 2). However, a lower number of students had extracurricular science experiences shared with family and friends (63%) or alone (53%). Students majoring in a STEM field in college had a higher rate of experiences with family or friends (68%) and alone (57%) relative to non-STEM majors (53% and 44%, respectively). Female students were more likely than male students to have had science experiences in social settings (67% to 54%), while there were no differences between genders when students were alone.

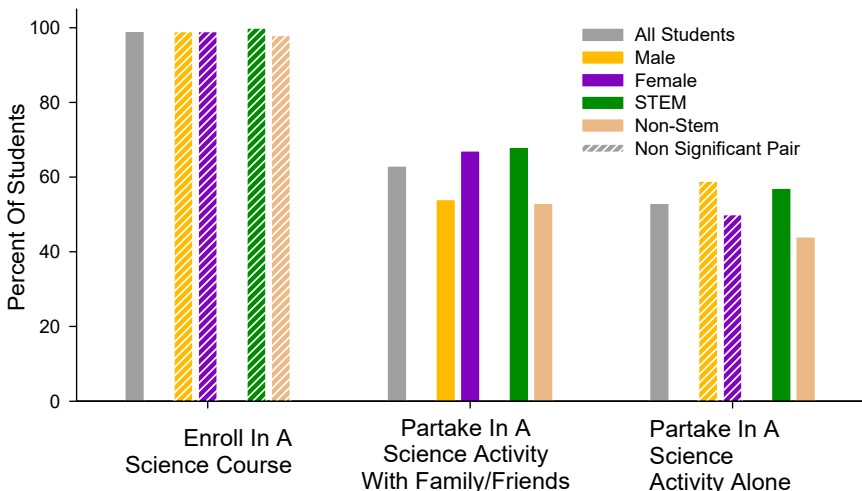

**Figure 2.** Student participation with science experiences. All students are shown (grey), together with the data split by self-identified gender (males/yellow and females/purple) and college major (STEM/green and non-STEM/tan). Non-significant differences between gender and major splits are shown for each experience type are shown as white hatched bars.

When asked if they enjoyed their pre-college experiences with science, results indicated differences between both genders and college majors (Figure 3). While the majority of students enjoyed science classes (84%), there was a predictably smaller number of non-STEM students (73%) that enjoyed science classes relative to STEM students (89%). The average student enjoyed science with family and friends (61%) more so than science alone (51%). As with science classes, STEM students were more likely to enjoy science with family and friends (66%) and alone (55%) compared to non-STEM students (52% and 42%). It should be noted that gender differences between science experienced with family or friends and alone switched along gender lines, with female students more likely to enjoy science socially (65% compared to male students at 53%) and male students more likely to enjoy science alone (58% compared to female students at 47%).

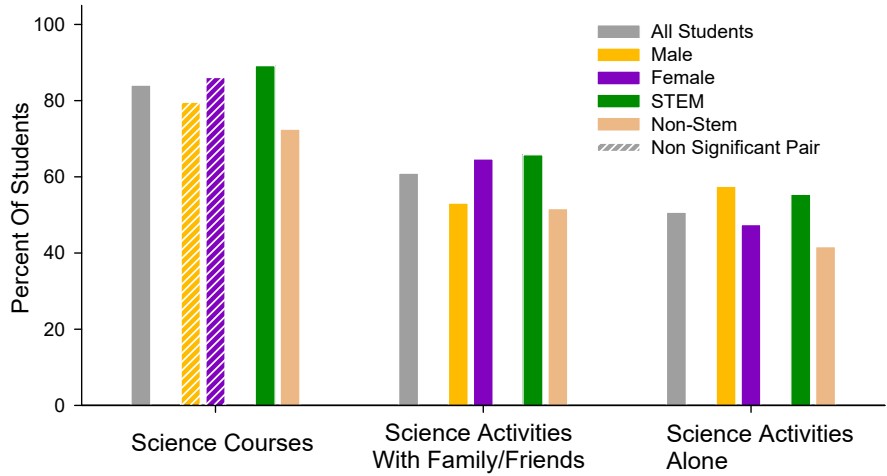

**Figure 3.** Student enjoyment of science experiences. All students are shown (grey), together with the data split by self-identified gender (males/yellow and females/purple) and college major (STEM/green and non-STEM/tan). Non-significant differences between gender and major splits are shown for each experience type are shown as white hatched bars.

When students were asked what percent of the time they had a positive experience in science classes (Figure 4), the results were an approximately normal distribution centered on 75% of the time. Both female students and STEM majors tended to have a higher percentage of positive experiences relative to male students and non-STEM majors.

**What Percent Of The Time Did You Have Positive Expereinces...**

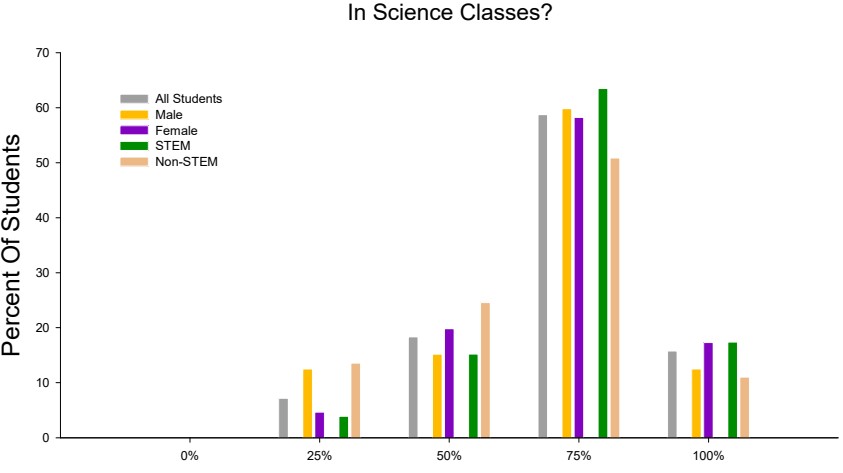

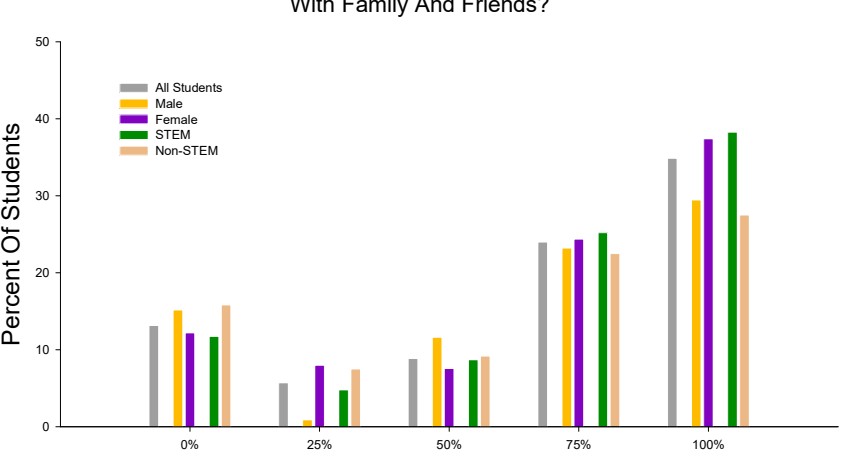

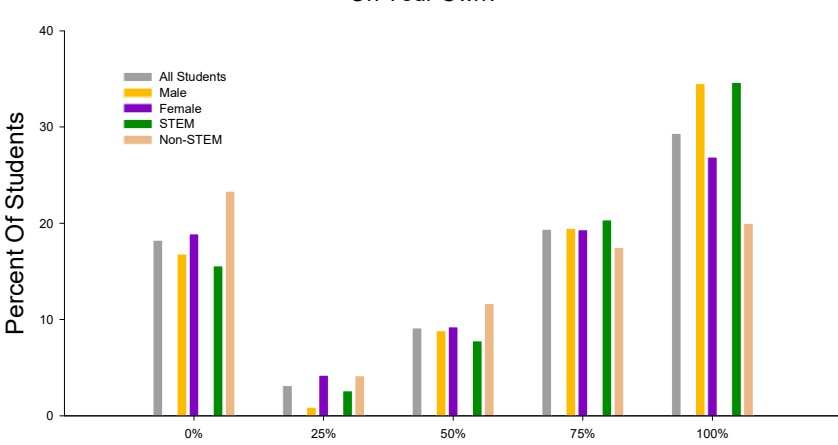

**Figure 4.** What percentage of time that students are having positive science experiences. All students are shown (grey), together with the data split by self-identified gender (males/yellow and females/purple) and college major (STEM/green and non-STEM/tan).

When asked what percent of the time students had a positive experience with science with friends or family and alone (Figure 4), the results were bimodal with a clear split between students primarily reporting positive experiences 0% of the time or 75–100% of the time. The average student had higher rates of positive experiences with family and friends relative to positive experiences alone. Results for male students were more bimodal for both family or friends and alone than any other population split, with nearly zero students reporting positive experiences 25% of the time. Compared to female students, male students were less likely to have positive experiences with family or friends and more likely to have positive experiences alone. Unlike the differences between genders, STEM majors had more positive experiences in both family or friend contexts as well as alone. The differences between STEM and non-STEM majors were large, particularly when reporting having positive experiences 100% of the time, with a gap of 11% for family or friends and 15% when alone.

Students were asked when they had their first positive and first negative experience with science in the classroom, with family and friends and when alone (Figure 5). Considering the most typical response, students had positive science experiences and higher rates of never having had a negative experience earlier in their education. In the classroom, every student had a positive experience at some point, as zero students reported never having a positive experience with science in the classroom. Positive experiences started in the elementary classroom, while first negative experiences were common in middle school. With family and friends, a slight majority of students (56%) had never had a negative experience, while again, the first positive experiences happened early. First negative experiences alone were similar to negative experiences with family/friends; however, a larger number of students report never having a positive experience alone (22%) compared to with family or friends (18%).

When comparing differences between genders in the classroom, the differences were in the timing of the first negative experiences, with few differences in the timing of positive experiences. Male students had more first negative experiences in elementary school and were more likely to report never having a negative experience, while female students were more likely to have had their first negative experience in middle school or college. With family and friends' female students had more first positive experiences, and male students had more negative experiences in elementary school, while male students had more positive experiences in middle school and female students were more likely to never have negative experiences with family and friends. There were no major differences between genders in the context of being alone, but female students did report higher rates of never having positive experiences with science alone.

The differences between STEM and non-STEM majors were large in the context of science experiences in the classroom, with non-STEM majors having more positive and negative experiences early and STEM majors having higher rates of never having a negative experience. With family and friends, STEM majors were more likely to have earlier positive experiences and more likely to never have had a negative experience, while non-STEM majors were more likely to never have had a positive science experience. When alone, non-STEM majors were more likely to have never had positive experiences.

When asked about their current comfort level with both science as a process and science content (Figure 6), most students reported their current comfort level with both the science process and science content as neutral, comfortable, or very comfortable. However, there were small differences between the groups of gender and major for science content because interest in science process varied between subgroups. More male students reported being very comfortable, while female students reported being only comfortable. Science majors were comfortable or very comfortable as well, regardless of gender.

# First Expereinces With Science
## In The Classroom

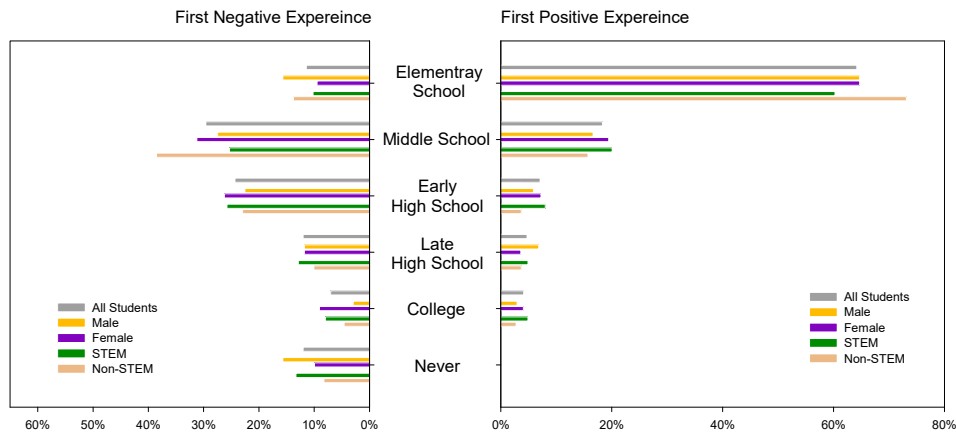

## With Family And Friends

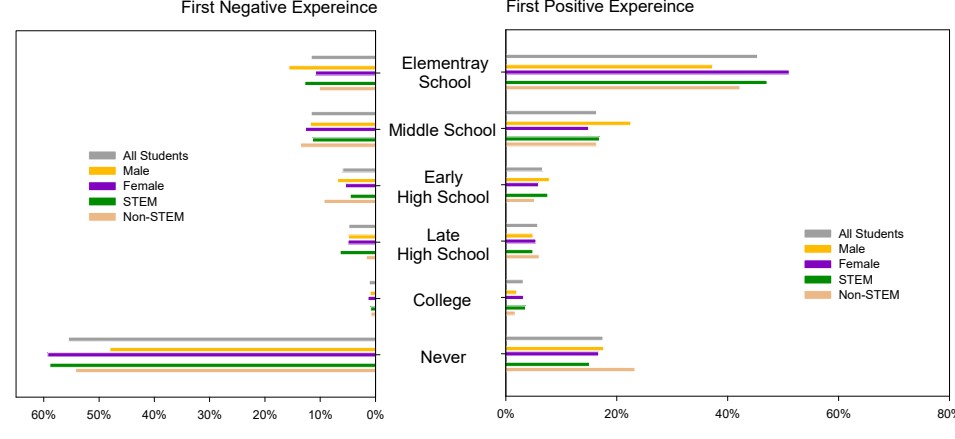

## Alone

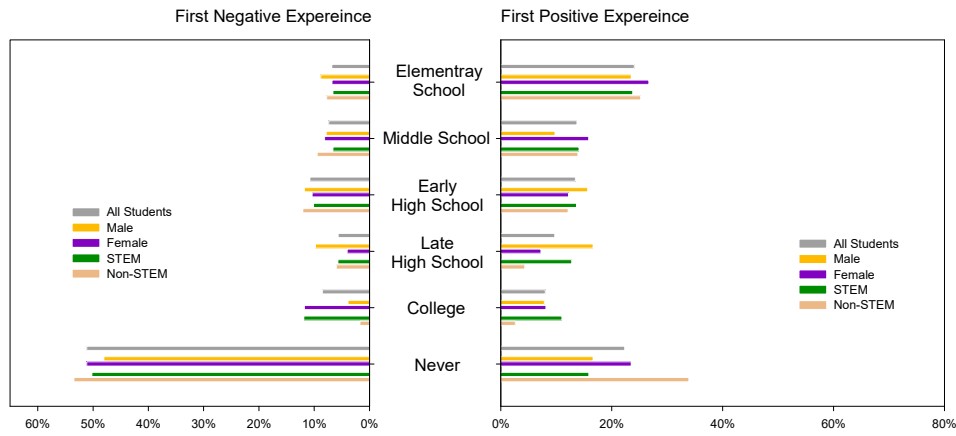

**Figure 5.** The timing of first positive (**right**) and negative (**left**) experiences. All students are shown (grey), together with the data split by self-identified gender (males/yellow and females/purple) and college major (STEM/green and non-STEM/tan). Non-significant differences between gender and major splits are shown for each period are shown as white hatched bars.

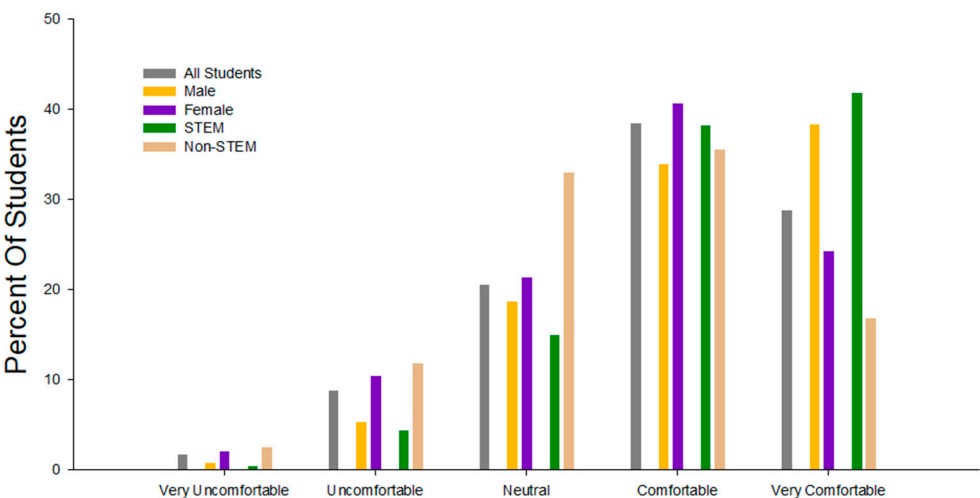

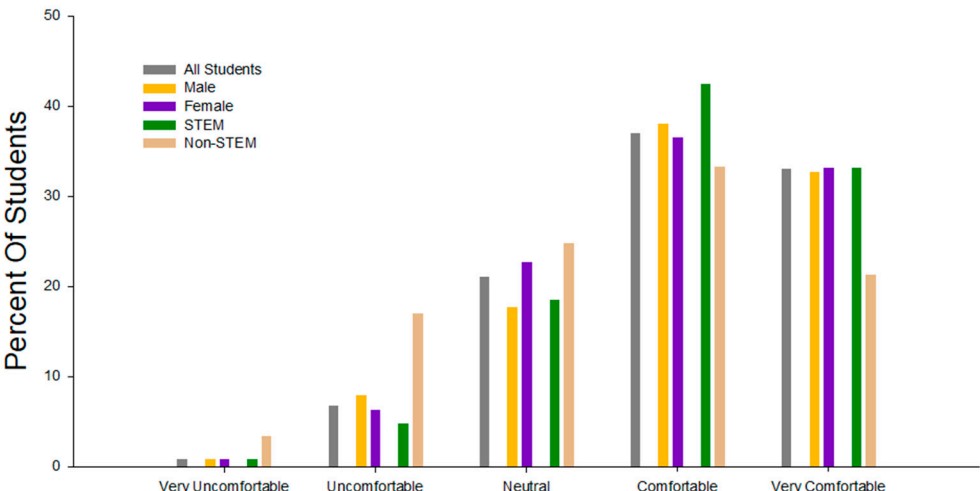

**Figure 6.** Current college-level student interest in both science as a process and science content is shown. All students are shown (grey), together with the data split by self-identified gender (males/yellow and females/purple) and college major (STEM/green and non-STEM/tan).

Students were also asked to report either positive or negative interest in multiple aspects of science (Figure 7). Results showed a gradient of positive interest across all aspects, but the mathematical and statistical aspects were clearly identified as areas with high amounts of negative interest (results not shown for all students). Female students had a more-negative view toward the mathematical ($p = 0.05$) and statistical ($p = 0.05$) aspects of science, compared to male students who more frequently enjoyed the mathematical ($p = 0.04$) and statistical ($p = 0.02$) aspects of science. However, female students enjoyed collaborative lab work more than male students did ($p = 0.05$). Most aspects did not have a statistically significant difference between male students and female students.

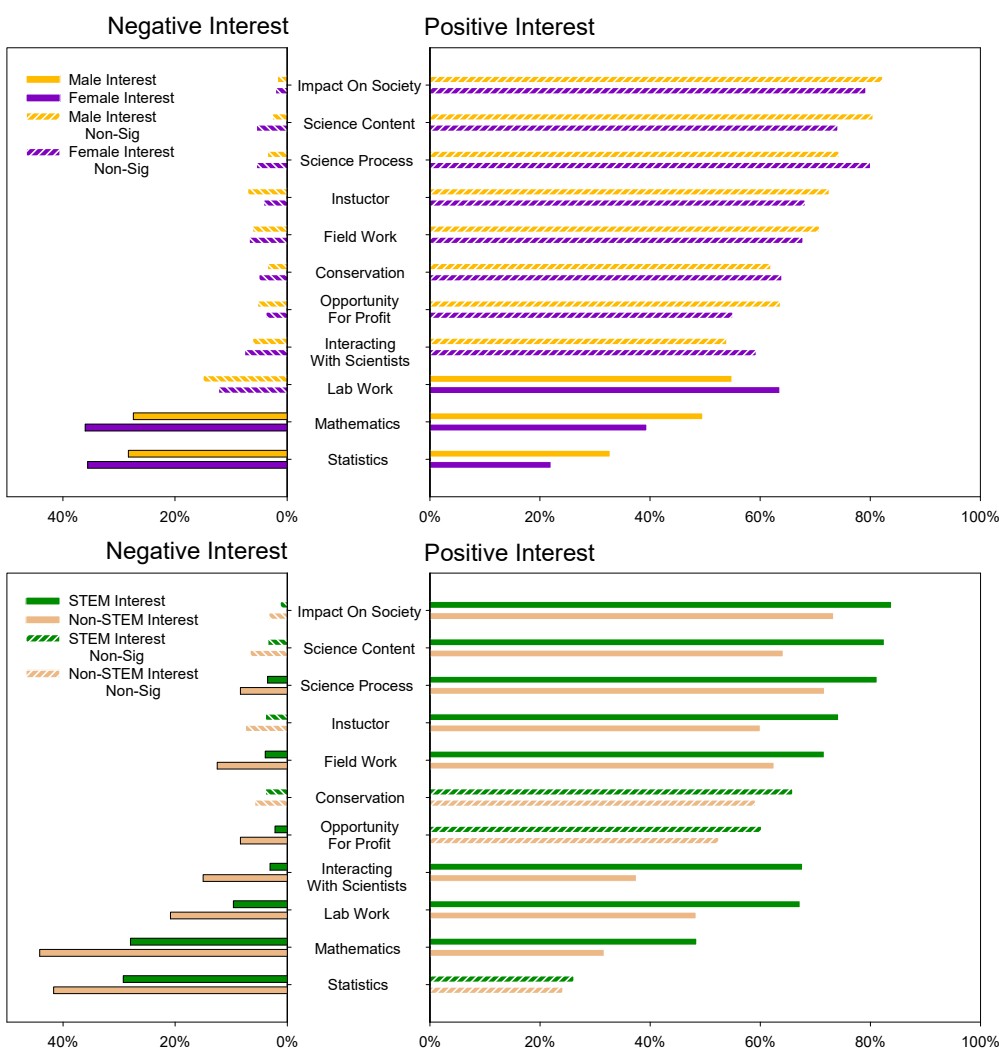

**Figure 7.** Current college-level student interest in multiple aspects of science content is shown. Data are split by self-identified gender (males/yellow and females/purple) and college major (STEM/green and non-STEM/tan). Non-significant differences between gender and major splits are shown for each period are shown as white hatched bars.

The difference in the enjoyment of the different aspects of science between STEM and non-STEM majors told a different story. The statistically significant negative interest ($p \leq 0.05$ in all cases) in aspects of science were all reported by non-STEM students (including science process, fieldwork, opportunities for profit, lab work, interacting with scientists, mathematics, and statistics). Contrarily, the statistically significant positive interest aspects were all from STEM majors and included impact on society, science content, science process, instructors you had, fieldwork, lab work, interacting with scientists, and mathematics.

## 4. Discussion

### 4.1. In-Classroom vs. Out-of-Classroom Experiences

Student experiences with science in the classroom have been the focus of many research reviews [7,10,32,33]. In most cases, gender is reported as a significant factor in how students think about science. This gender effect has been found to be variable among science disciplines, with male students having a larger positive attitude towards physics than biology [7]. However, attitudes and achievement in science are not tightly correlated to each other, with some studies showing a correlation between 0.3 and 0.5 and some studies

showing no link at all between attitudes and achievement [33]. With an increasing need for a society engaged with science, focusing on the gendered science achievement gap should not be the only big-picture goal.

Our results showed no significant differences between genders in student enjoyment of science courses or the timing of the first positive science experience in the classroom. However, male students were more likely to have had their first negative classroom experience in elementary school or never have had a negative classroom experience, while female students were more likely to have had negative experiences when mathematics was included in the science curriculum. It is also possible that gender differences reported in high school can be caused by separating general science courses into field-specific courses during that time, with separate classes for physics, biology, and chemistry. Gender differences in attitude towards science based on science type (physics, biology, or chemistry) are noted in the meta-analysis of Weinburgh [7], with both genders having a positive correlation in biology and physics courses, and that correlation is stronger for female students than male students.

When considering potential ways to close the gender gap, Kerr and Robinsthon Kurpius [34] discussed focusing on increasing self-esteem and self-efficacy in female STEM students. With every student having positive experiences in the classroom, and the majority of those being early, working to increase self-esteem by supplying more positive experiences might not be the best method. The work of Brownlow et al. [27] connects increased levels of science anxiety to student self-reporting of non-helpful teachers. Thus, focusing on reducing negative experiences such as unclear learning goals, unclear communication, poor contextualization of science concepts, or an overall unsupportive learning environment might help female students build confidence early on, helping to close the gender gap in the sciences.

Interestingly, students not majoring in STEM fields have positive as well as negative first experiences with science earlier than other students. It is possible that having earlier positive experiences would benefit the odds of later ending up in STEM fields, and negative experiences seem to be more influential than positive because of the correlation with a disinterest in science later in life. In elementary school, when science is qualitative, negative experiences can be attributed to problems with science concepts or the student's instructor [9,14]. Having problems early on with concepts will not self-correct itself when quantitative math and statistical concepts are added to science later in life. Again, Brownlow et al. [27] explain that teachers and their pedagogy are key to limiting science anxiety. However, interest in science is not only fostered within classroom walls, and out-of-classroom experiences have a role to play as well.

Expectedly, nearly 100 percent of students report having been enrolled in a science class before college. While the in-class curriculum experienced by many students surveyed may have been similar, given the relatively close location of colleges, the extracurricular science activity experiences were not identical for students. Research from Kahle et al. [35] suggests that while students have similar classroom experiences, male students typically have more extracurricular science activities at the high school level. Having positive experiences outside of the classroom both increases the enjoyment of science and reduces anxiety levels because students can learn scientific concepts and applications in low-stress situations with peers and away from teachers and grades.

The recent work of Stoet et al. [36] highlights a multi-generational societal effect of negative science experiences. They showed that levels of math anxiety were higher in more developed countries, and differences between genders were more pronounced in more developed countries as well, with female students having more negative experiences and high anxiety related to mathematics. However, female student anxiety levels were not connected to the amount of engagement of their mothers in STEM fields, showing that having a STEM role model of the same gender in the house would not remove gender differences in the following generation. Our results highlight higher rates of science engagement with family or friends with female students, as well as higher rates of enjoyment

of those experiences with family or friends are related to positive perceptions of science. However, those are positive experiences, and overall engagement with science needs to be thought of as a summation of both positive and negative experiences. Having higher amounts of enjoyment from positive science experiences might not lead to more female students in STEM fields in the long run if negative issues, such as higher amounts of math anxiety, are not addressed.

Having students engaged socially in science (with family or peers) happens at a higher rate for female students than male students. Having people around with similar interests and passions may be an insulating barrier from the strong negative social stigma about female students with STEM interests. Similar to results from in-classroom research [34], focusing on female students' self-esteem and self-efficacy in social settings away from the classroom would be an effective strategy to help close the STEM gender gap.

Students not majoring in STEM fields in college were significantly more likely to never have had a positive experience outside of the classroom, either socially or alone. These results are compatible with the work of Ainley and Ainley [37], showing that early adolescent enjoyment of science is not dependent upon knowledge of science but is linked to continuing interest in science in the future. Students that do not end up majoring in STEM fields have lower rates of participation and enjoyment of science outside of the classroom. This lack of enjoyment and participation can be thought of as a chicken-and-egg situation in elementary and middle school ages, but after enough negative experiences, their interest in science lags behind students who had fewer negative experiences and may later go on to major in STEM fields.

### 4.2. Self-Identified Gender in STEM Fields

The problem of female student attrition in STEM fields at both the K-12 and college level is well studied. The idea of a "leaky pipeline" wherein women lose interest well before reaching college is a complex problem with many interactions [38]. Female students show high levels of interest early in their K-12 careers, but male student interest passes these levels, later on, demonstrating the "leaky pipeline" in action.

For women who do not lose interest in STEM fields, their interests appear to focus on different areas within STEM fields than do male students' interests. While many students may have negative feelings towards the mathematical and statistical aspects of science, these feelings seem to be more frequently observed among female students [36]. This likely results in female students gravitating towards fields with less on-the-job use of mathematics or statistics. For example, Sadler et al. [12] reported female students expressing more interest in health and medicine careers in high school relative to male students, who typically expressed more interest in engineering fields.

While both Blickenstaff [38] and Sadler [12] presented results on interest levels in science between genders, the link between high-interest level and continuance in STEM may not be supported by research (i.e., The congruence problem in John Holland's theory of vocational decisions [39]). Instead, some propose the issue lies not directly with interest but, instead, confidence. As a woman's confidence level in her ability to do science [17] or engineering [40] drops, retention rates also decline. This loss of self-confidence may be due to hostility within the field that women encounter from faculty, instructors, or peers [41,42]. It is interesting to note that outside of differences in math/statistics interests, female students reported being more interested in lab work. We suggest that this speaks to the perception of the importance of the collaborative nature of science, which is more common in female students [43].

### 5. Conclusions

Students enter college with a myriad of pre-determined attitudes about science that are rooted in their pre-college experiences with science. These are a mix of positive and negative experiences as well as formal and informal experiences. Our survey results demonstrate that students steadily gained interest in science from elementary school to college, but there

is little difference in pre-college interest as it relates to specific science content compared to the process of doing science. In the early years of college, students show greater comfort in science content compared with lower comfort levels with the scientific process. As a whole, students show a marked decrease in positive interest and a corresponding increase in negative interest in quantitative aspects of science (mathematics and statistics) while generally feeling positive towards all other aspects of science. While every student reported taking science classes before entering college, the majority of students also have out-of-the-classroom experiences, both with family and friends as well as alone. While not every student enjoyed their science experiences, a majority of students reported positive experiences before college. When looking at the first positive and negative experiences with science, student positive experiences were reported earlier in life than first negative experiences, and when asked about first negative experiences with family and friends, simply having negative experiences at all was uncommon. These results highlight the importance of building science experiences out of the classroom, as enjoyment leads to continual and future interest in science [30].

The differences between self-identified genders were apparent, with female students having more positive experiences with science in social settings and male students preferring science alone. Female students reported starting with a high interest in science but ended up being surpassed by male students during high school. There were also differences in gender in current comfort level in understanding the scientific process, with female students being less comfortable than male students. The same and greater effects were seen in female students where elementary school interest in science was high very, but instead of growing slowly, it declined over time. One potential way to help close gender gaps in STEM fields is to help young female students to have positive, self-esteem-building experiences with family or peers outside of the classroom. Helping to create social groups where negative stigmas about science are not present could be a strong and simple way to help female students grow their passion for science early in life.

This work highlights the importance of outreach by scientists toward the next generation of scientists, both formally and informally. This outreach can be an overlooked part of a scientist's job, and both research on outreach effectiveness and resources to assist in outreach are available [44–46]. For students of any age, out-of-the-classroom experiences are important in creating and improving attitudes towards science.

**Supplementary Materials:** The following supporting information can be downloaded at: https://www.mdpi.com/article/10.3390/educsci12070496/s1.

**Author Contributions:** Conceptualization, D.E.R., D.A.M. and R.A.J.; methodology, D.E.R.; formal analysis, E.C.K. and D.E.R.; writing—original draft preparation, D.E.R. and E.C.K.; writing—review and editing, D.E.R., E.C.K., D.A.M. and R.A.J.; funding acquisition, D.E.R. All authors have read and agreed to the published version of the manuscript.

**Funding:** This work was supported by Dickinson College's Center for Sustainability Education (#RA152) and the National Science Foundation Atmospheric and Geospace Sciences Postdoctoral Fellowship Program (#GEO-1430396).

**Institutional Review Board Statement:** The study was conducted in accordance with the Declaration of Helsinki and approved by the Institutional Review Board of the University of Oklahoma (2016).

**Informed Consent Statement:** Informed consent was obtained from all subjects involved in the study.

**Data Availability Statement:** Due to the research data being from human subjects, data from this study are not publicly available.

**Acknowledgments:** We would like to thank Leanne May and Tony Yates for helping us administer the survey and Gordon Uno for assistance in creation of the survey. The authors would like to thank Yost R. for dogged support with the database management.

**Conflicts of Interest:** The authors declare no conflict of interest.

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
