# Peer review of "Student Experiences and Changing Science Interest When Transitioning from K-12 to College"

_education, doi:10.3390/educsci12070496_

Round 1

Reviewer 1 Report

The article is very interesting and deals with a relevant topic in the didactics of experimental sciences. I would like to congratulate the authors. It is very well written and with relevant references. I just have a few issues that I think should be addressed:

1) in the introduction, Science literacy in formal settings is developed, but given the nature of the article, it should describe more sciece literacy in informal settings. 

2) In Materials and Methods: it is recommended to offer more information about the design of the survey used: validation process, number of questions, or when it was supplied .At the ethical level, specify if the participants were informed before doing the survey. And if it was done prior to the start of the academic year.

3) It is suggested to change the color code for the figures (do not put pink for females and blue for males).

4) It is recommended to review references, some are detected in capital letters.

Reiterate again my congratulations to the author for the article.

Author Response

The article is very interesting and deals with a relevant topic in the didactics of experimental sciences. I would like to congratulate the authors. It is very well written and with relevant references. I just have a few issues that I think should be addressed:

            We would like to thank Reviewer 1 for their time and comments. Our responses are below.

1) in the introduction, Science literacy in formal settings is developed, but given the nature of the article, it should describe more sciece literacy in informal settings.

            In conjunction with Reviewer 2’s comments, we’ve worked to add more information on informal science literacy on Lines 57-59

2) In Materials and Methods: it is recommended to offer more information about the design of the survey used: validation process, number of questions, or when it was supplied .At the ethical level, specify if the participants were informed before doing the survey. And if it was done prior to the start of the academic year.

            We have provided this information on the survey on Lines 117-122 and also included the survey as a Supplementary File.

3) It is suggested to change the color code for the figures (do not put pink for females and blue for males).

            This is a good suggestion and we’ve changed to blue/pink color code to yellow/purple

4) It is recommended to review references, some are detected in capital letters.

            We have worked to clean the reference list up

Reiterate again my congratulations to the author for the article.

            Thank you!

Reviewer 2 Report

The overall manuscript is neat and written concisely—with relevant information for existing literature. I have listed a few small improvements (e.g., focused on clarification and consistency).

Author Response

The title as well as the introduction raised expectations about your manuscript and research.

The topic you are addressing would be a relevant addition to existing literature. Thank you for

this valuable contribution. I will structure my feedback in (a) general remarks (these

comments cover feedback applicable in the entire manuscript), and (b) specific remarks

(feedback on sentence and/or word level). The specific remarks can include a quote from your

original manuscript to refer to a specific section. The specific remarks will refer to page

(emphasis added in boldface; e.g., 1.15/16) and row(s; e.g., 11.15/16).

General remarks:

The overall manuscript is neat and written concisely with relevant information for existing

literature. I have listed a few small improvements (e.g., focused on clarification and

consistency).

            We thank Reviewer 2 for their time and detailed comments. The edits we’ve made following your comments have clearly improved the manuscript.

Specific remarks:

p.1.6 You mention genders. Nowadays you need to elaborate on that. I have noticed

that you focus on male/female only. Can you explain to me why you only focus

on these two?

            This is a very valid comment and to be honest, this work was planned in 2015 and conducted in 2016. If we were conducting this survey today, we would handle this issue differently. In our survey we asked students to self-identify their gender between Male and Female which is hindsight was limiting. However, there has also been decades of research in this area which would be unfair to disregard.

            To attempt to address this issue, we’ve changed to the wording here and throughout the manuscript to “self-identified gender”. We’ve added the passage on Line 49-50 to introduce gender that readsWith noting that the idea of gender is broad, complex, and can change, student gender has long been studied …” Based on comments here and on comments from Reviewer 1 we’ve changed the blue/pink color scheme to yellow/purple. If there is a better way to handle this issue, we are happy to continue to edit this manuscript further.

p.1 I find it bold to immediately introduce the transfer from K-12 to college. This

should be presented with more nuances.

            We introduce this idea halfway through the Introduction section, on Line 65, but to keep the reader we have added additional text and citations at that section.

p.1.13 when alone = by themselves? When alone is an uncommon way of describing that.

            In the survey we asked students about their past experiences with science 1) “in the classroom”, 2) “with family and friends”, and 3) “on your own”. We have edited this line for readability.

p.1.39 This sentence requires sources.

            We’ve added an additional citation to this line

p.2.57 Use and instead of & .

            We have made this change

p.2.83/84 You capitalise the first letter after (2) but not after (1). Please revise this to

make it consistent.

            We have made this change

p.2.86 How do you conceptualise views ? Interests in? Perceptions of? Understanding? If this is what you mean, you need to introduce this. Because you speak about interest and understanding (p.3.100).

            We have edited this line

p.2.89 While not without limitations Does not belong here. You can mention this

in the discussion.

            We have removed this section of text.

p.3.105 The n needs to be placed in italics. Also go over the use of spaces with these

(it is inconsistent). In a similar vein, the p needs to be placed in italics as

well.

            We have edited the manuscript through to fix this issue

p.3.129 The word very is redundant. You need to propose a valid argument instead.

            We have edited this line and we use this transition line to setup the second half of this paragraph.

p.4.Figure1 I would give the bars patterns (like you did with the non-significant pairs).

            Could you be more clear what you’re suggesting here? We are working to show all the data, including non-significant differences. Are you suggesting that the significant pairs also have a pattern to them? We have color colors to show significance in all of our figures.

p.4.148 What does the / mean? It is and, or, or and/or? Also clarify this in the

remainder of your work. If your Figure 4 you write it as Family and Friends .

            We’ve made this change throughout the manuscript

p.7.Figure4 You do not explain what the X-axis means.

            We explain this in the Figure title, but we’ve added this information to the caption as well

p.8.196 average responses = vague. Can you clarify this?

            We have edited this line for clarity

p.8.209 more likely to never have had a negative experience I am wondering if this is true. I have issues with the label never .

            When asked when their first positive or negative experience was, “never” was a possible survey answer. That is what we are reporting here. We’ve edited this line to “… more like report never having a…”

p.11.249 252 The p need to be placed in italics.

            We have made this change.

p.11.255 The <= needs to be replaced by the.

            We have made this change.

p.12.275 You describe the whole society . What is the difference with society ? I do not see the use of adding the word whole.

            We have edited this line

p.12.286 The 18 study is redundant.

            We have edited this line

p.12.294 297 But these are also relevant for males (this is not gender-only). The issues your

describe are aspects that need to be tackled in education.

            Yes, but this paragraph is connecting to the idea of “closing the gender gap”. We have edited to line to make that point clear.

p.12.299 While on its face = informal. Can you come up with a more formal alternative?

            We have edited this section of text.

p.12.303 This statement needs a source.

            We have added two citations to this line

p.12.305 Concepts such as?

            We have edited this line to read “quantitative math and statistical concepts”

p.12.309/310 But science is mandatory in the curriculum.

            It is mandatory for students in the USA, but not necessarily true outside of the USA. And, we use this statement as a way to introduce the topic of “in class experiences”

p.13.353 The word complex is redundant (in many complex interactions ).

            We have edited this line

p.13.365 Earlier you used the word gender(s) instead of sexes. It might be wise to switch

to sexes, because that implies male/female whereas genders has a more complex conceptualization.

            We have standardized the text, using the term gender

p.14.367 some propose Who does? Such as?

            We have making a reference to the proceeding sentence, were we have two citations on this topic.

p.14.407 415 This section does not belong here. You propose new literature. Embed this in

the introduction.

            We have cut this section.

p.references Use an en dash instead of a hyphen between the page numbers. Make the

capital letter use in the title consistent. I also spot several et al . Please insert

the remaining authors.

            We have worked to clean up the reference section.

Reviewer 3 Report

First of all, I would like to thank the authors for their contribution "Student experiences and changing science interest when transitioning from K-12 to college".

I am generally very sympathetic toward the project of this paper. 

While Ι found great value in your paper, I feel that it will take a bit more time and effort to make it suitable for publication.

Furthermore, the reference list is a little bit weak, author(s) need to strengthen their theoretical framework. I propose some of  the following articles:

Ampartzaki, M., Kalogiannakis, M., Papadakis, St., & Giannakou, V. (2022). Perceptions About STEM and the Arts: Teachers’, Parents’ Professionals’ and Artists’ Understandings About the Role of Arts in STEM Education. In St. Papadakis & M. Kalogiannakis (Eds), STEM, Robotics, Mobile Apps in Early Childhood and Primary Education - Technology to promote teaching and learning. Lecture Notes in Educational Technology, (pp. 601-624). Switzerland, Cham: Springer, https://doi.org/10.1007/978-981-19-0568-1_25

Kanaki, K., & Kalogiannakis, M. (2022). Assessing Algorithmic Thinking Skills in Relation to Age in Early Childhood STEM Education. Education Sciences, 12(6), 380. https://doi.org/10.3390/educsci12060380

Kanaki, K., & Kalogiannakis, M. (2022). Assessing algorithmic thinking skills in relation to gender in early childhood, Educational Process, 11(2), 44-59.

Alexandre, S., Xu, Y., Washington-Nortey, M., & Chen, C. (2022). Informal STEM Learning for Young Children: A Systematic Literature Review. International Journal of Environmental Research and Public Health, 19(14), 8299.

Wang, M. T., & Degol, J. L. (2017). Gender gap in science, technology, engineering, and mathematics (STEM): Current knowledge, implications for practice, policy, and future directions. Educational psychology review, 29(1), 119-140.

Jackson, M. C., Leal, C. C., Zambrano, J., & Thoman, D. B. (2019). Talking about science interests: the importance of social recognition when students talk about their interests in STEM. Social Psychology of Education, 22(1), 149-167.

Overall, the research process and the results could be explained more thoroughly to make the study more transparent and informative. I encourage you to more fully illuminate your analysis process. 

Author(s) need to mention ethical issues for their study.

I wish you the best of luck with the revisions of your manuscript.

Author Response

First of all, I would like to thank the authors for their contribution "Student experiences and changing science interest when transitioning from K-12 to college".

I am generally very sympathetic toward the project of this paper.

While Ι found great value in your paper, I feel that it will take a bit more time and effort to make it suitable for publication.

            We thank Reviewer 3 for their time and comments on our work and we’ve worked to address your comments below.

Furthermore, the reference list is a little bit weak, author(s) need to strengthen their theoretical framework. I propose some of  the following articles:

            We have worked to add as many as these citations as we can into our manuscript. Thank you for the assistant in this broad field of literature.

Ampartzaki, M., Kalogiannakis, M., Papadakis, St., & Giannakou, V. (2022). Perceptions About STEM and the Arts: Teachers’, Parents’ Professionals’ and Artists’ Understandings About the Role of Arts in STEM Education. In St. Papadakis & M. Kalogiannakis (Eds), STEM, Robotics, Mobile Apps in Early Childhood and Primary Education - Technology to promote teaching and learning. Lecture Notes in Educational Technology, (pp. 601-624). Switzerland, Cham: Springer, https://doi.org/10.1007/978-981-19-0568-1_25

Kanaki, K., & Kalogiannakis, M. (2022). Assessing Algorithmic Thinking Skills in Relation to Age in Early Childhood STEM Education. Education Sciences, 12(6), 380. https://doi.org/10.3390/educsci12060380

Kanaki, K., & Kalogiannakis, M. (2022). Assessing algorithmic thinking skills in relation to gender in early childhood, Educational Process, 11(2), 44-59.

Alexandre, S., Xu, Y., Washington-Nortey, M., & Chen, C. (2022). Informal STEM Learning for Young Children: A Systematic Literature Review. International Journal of Environmental Research and Public Health, 19(14), 8299.

Wang, M. T., & Degol, J. L. (2017). Gender gap in science, technology, engineering, and mathematics (STEM): Current knowledge, implications for practice, policy, and future directions. Educational psychology review, 29(1), 119-140.

Jackson, M. C., Leal, C. C., Zambrano, J., & Thoman, D. B. (2019). Talking about science interests: the importance of social recognition when students talk about their interests in STEM. Social Psychology of Education, 22(1), 149-167.

Overall, the research process and the results could be explained more thoroughly to make the study more transparent and informative. I encourage you to more fully illuminate your analysis process.

            We have worked to add additional information about our analysis in the Methods section

Author(s) need to mention ethical issues for their study.

            Following this comment and a similar comment from Reviewer 1, we have added this information to the Methods section.

I wish you the best of luck with the revisions of your manuscript.

            Thank you!